# Development of a Nomogram to Predict the Outcome for Patients with Soft Tissue Sarcoma

**DOI:** 10.3390/vetsci10040266

**Published:** 2023-03-29

**Authors:** Jonathan P. Bray, John S. Munday

**Affiliations:** 1AURA Veterinary, 70 Priestley Road, Surrey Research Park, Guildford GU2 7AJ, UK; 2School of Veterinary Science, Massey University, Palmerston North 4410, New Zealand

**Keywords:** nomogram, prognosis, soft tissue sarcoma, immunohistochemistry

## Abstract

**Simple Summary:**

Local recurrence following surgical resection is one of the most common causes of treatment failure and tumour-related death for dogs with soft tissue sarcoma. Currently, the clinician will consider the characteristics of the individual patient and tumour to decide whether adjuvant therapy or further surgery should be considered to reduce the risk of recurrence following surgery. However, this risk assessment is subjective and influenced by clinician bias, meaning some patients will undergo additional treatments that may be unnecessary, while other patients will miss out on potentially life-saving treatment. A nomogram is a simple-to-use statistical device that allows the clinician to calculate an objective risk measurement from a complex algorithm that incorporates a number of important patient and tumour characteristics. In this study, a nomogram was developed to predict the risk of recurrence in a cohort of dogs previously treated for STS. The nomogram developed in this study accurately predicted tumour-free survival in 25 patients but failed to predict recurrence in 1 patient. The current study provides the first evidence in veterinary oncology to support a role for the nomogram to assist with predicting the outcome for patients after surgery for STS.

**Abstract:**

Soft tissue sarcomas (STSs) are common cutaneous or subcutaneous neoplasms in dogs. Most STSs are initially treated by surgical excision, and local recurrence may develop in almost 20% of patients. Currently, it is difficult to predict which STS will recur after excision, but this ability would greatly assist patient management. In recent years, the nomogram has emerged as a tool to allow oncologists to predict an outcome from a combination of risk factors. The aim of this study was to develop a nomogram for canine STSs and determine if the nomogram could predict patient outcomes better than individual tumour characteristics. The current study provides the first evidence in veterinary oncology to support a role for the nomogram to assist with predicting the outcome for patients after surgery for STSs. The nomogram developed in this study accurately predicted tumour-free survival in 25 patients but failed to predict recurrence in 1 patient. Overall, the sensitivity, specificity, positive predictive, and negative predictive values for the nomogram were 96%, 45%, 45%, and 96%, respectively (area under the curve: AUC = 0.84). This study suggests a nomogram could play an important role in helping to identify patients who could benefit from revision surgery or adjuvant therapy for an STS.

## 1. Introduction

Soft tissue sarcomas (STSs) are common cutaneous or subcutaneous neoplasms in dogs. Currently, most STSs are initially treated by surgical excision. While many STSs are able to be cured by this method, local recurrence may develop in almost 20% (range 7–75%) of patients, with recurrence consistently associated with reduced overall survival for the dog [1,2]. Currently, it is difficult to predict which STS will recur after excision, but this ability would greatly assist patient management. This could allow a clinician to decide whether an individual patient is likely to benefit from adjuvant chemotherapy or radiotherapy [3], or whether additional surgery should be performed to remove residual tumour from the wound bed [4].

The benefit of accurately predicting tumour behaviour is not limited to canine STSs; numerous methods have been previously developed to help clinicians predict the prognosis for a patient with many different types of cancer [2]. Historically, the gold standard for prognostication in human oncology is considered to be the tumour, node, and metastasis (TNM) system [5,6]. This system has been described for most forms of human cancer since 1953 [6] and was first applied to veterinary oncology in 1980 [7]. A TNM system has been described for canine STSs, but this has not been validated in a clinical setting [8]. However, for STSs, the TNM system is unlikely to be highly prognostic as it assumes prognosis is determined by the presence of nodal or distant metastasis, which is at odds with the clinical reality of this disease. While metastatic spread will occur in a proportion of dogs with STSs, it is recognised that the majority of dogs that die from STSs are euthanised because of the local impact of their disease rather than the development of metastasis [9]. Additionally, the TNM system that determines prognosis is partially based on the tumour size. Tumour size is a valuable metric in humans, who are all roughly equal size. However, while tumour size has been found to predict prognosis in some studies of canine STSs [10,11], size is less useful in dogs due to the highly variable size of dogs of different breeds. In addition to size, other criteria that have been reported as prognostic include tumour grade, mitotic index, and the percentage of tumour necrosis [12,13].

Traditionally, a clinician uses their knowledge of the oncology literature, combined with their own experiences, to help determine the potential prospects for an individual patient with cancer. For example, an experienced veterinarian will know that the risk of recurrence for an STS larger than 5 cm is almost double the risk of a 1 cm tumour [2]. A high-grade STS also has a much higher risk of recurrence compared to a low-grade STS. Recurrence risks have been published for a range of tumour characteristics, including whether a resection margin is histologically clean, as well as the results of the histologic, immunohistochemical, or molecular analysis of the tumour. The challenge for a clinician is that a patient with cancer will present with a unique combination of tumour characteristics. The clinician needs to consider the combined influence of each independent tumour variable if they are to provide the patient with a realistic prognosis. The subjective nature of this decision may lead to some patients being under- or over-treated for their cancer. The nomogram allows a clinician to incorporate a number of clinical and tumour characteristics that are known to be significant to recurrence and provides a more objective index with which to support a treatment decision for an individual patient [14].

A nomogram (also called a nomograph, alignment chart, or abaque) enables the computation of a mathematical function using a simple graphical interface [15]. Before the age of calculators and computers, the nomogram was considered a vital tool for engineers working in a variety of industries, including aeronautics, railway construction, and the military. In recent years, the nomogram has re-emerged in the medical field as a potential tool to help patients and doctors to derive an accurate individual risk assessment for patients with a variety of conditions, including cancer [14].

In human oncology, nomograms have been developed for a variety of tumour types and clinical situations. For example, nomograms have been developed to estimate survival outcomes [15,16], the benefit of adjuvant therapy [17,18], the impact of a particular treatment on quality of life [19,20], and the risk of tumour recurrence [21]. Nomograms have been developed to determine the risk of a patient having an incomplete resection if a conservative surgical strategy is employed [22], or to identify patients who should have more extensive surgery or undergo nodal excision [23,24]. When determining the risk of cancer progression for an individual patient, some nomograms have been shown to be more reliable than the clinical judgement of the specialist clinician [25].

To date, nomograms have not been utilised in veterinary medicine to support clinical decisions. Therefore, the aim of the present study was to develop a nomogram for canine STSs and determine if the nomogram could predict patient outcomes better than the currently used individual tumour characteristics.

## 2. Materials and Methods

### 2.1. Patient Data

Separate datasets were used to develop the two nomograms created in this current study. The first dataset was derived from a series of 350 STS [26]. This dataset was called “Clinical”. Because accurate nomogram construction requires no missing or incomplete variables, 180 cases had to be excluded leaving 170 STS in the series. The second dataset utilised the STS that had been previously used in a previous study that examined the potential for using VEGF and decorin immunostaining to predict prognosis [27]. This dataset was termed “IHC” and contained 82 tumours. Because both the clinical and IHC cohorts were derived from the same parent population, some cases were utilised in both datasets.

To allow development of well-calibrated and validated nomograms, each model is ideally built using a training cohort of data and then validated against an independent validation cohort [28]. To establish these two required cohorts, the CRAN package “sampling” in R (R version 3.5.1, R Foundation for Statistical Computing, Vienna, Austria) was used to randomly select 68 cases from the Clinical dataset, representing 40% of the total available cases. These selected cases were used to create the validation cohort, called “Clinical_validate”. The cases remaining now created the larger training cohort, which consisted of 102 cases; this dataset was renamed “Clinical_train”. 

Because of the smaller number of cases in the IHC database, it was not possible to separate the dataset into two and still retain a meaningful number of events within each cohort. For this reason, it was not possible to create an independent cohort for the IHC nomogram to permit internal validation.

### 2.2. Patient Demographics and Risk Analysis of Individual Variables

All statistical analyses were performed with SPSS (IBM SPSS Statistics for Windows, Version 25.0. IBM Corp, Armonk, NY, USA). Local recurrence of the tumour within 3 years was the defined endpoint for the study. The disease-free interval (DFI) was defined as the time from surgery to the time when recurrence was identified by the referring vet. Patients were censored if they had died prior to the endpoint of the study and no tumour recurrence had been noted at that time, based on clinical records of the referring veterinarian.

The Kaplan–Meier method was used to compare DFI according to age, palpable characteristics, tumour size, histological characteristics (i.e., differentiation, necrosis, mitotic score, grade), and the development of local tumour recurrence. Finally, Cox regression analysis was performed to identify the categories of significance and their hazard ratios for patients whose tumours recurred within 3 years of surgery. A value of *p* < 0.05 was considered significant.

### 2.3. Using a ROC Curve to Evaluate the Predictive Accuracy of Individual Tumour Characteristics

For each category showing significance with Cox regression analysis, the test result was plotted against actual tumour recurrence in a receiver-operating-characteristic (ROC) curve. Using co-ordinates from the ROC curve, a cut-off value for 3-year local recurrence probability was determined by calculating the positive differential rate using the following formula: [sensitivity − (1-specificity)] [29]. This allowed for the determination of a probability value that provided an optimal balance of sensitivity and specificity. This enabled a binary recurrence outcome (i.e., yes or no) to be predicted based on the actual test result. By comparing this predicted outcome with the actual outcome in a 2 × 2 table, it was possible to calculate Sensitivity, Specificity, Positive Predictive Value and Negative Predictive Values for both the “Clinical_train” and the “IHC” nomograms. 

### 2.4. Nomogram Construction

To identify the independent predictors of time-to-event outcome that should be used in nomogram construction, multivariable Cox regression analysis was performed on all recorded clinical variables in the “Clinical_train” dataset, including age, the size of the tumour, palpable characteristics, location, as well as histological characteristics of the tumour including grade, differentiation, necrosis, mitoses and mitotic rate. A backward selection of variables was performed to obtain the model with the best fit. Due to the small size of the dataset, variables were selected for use in the model if their *p*-value was <0.15. 

Following the selection of the independent variables to be used in the model, nomograms were constructed using the ‘rms’ and ‘survival’ packages available in R (R version 3.5.1, R Foundation for Statistical Computing, Vienna, Austria), as described by Harrell [30]. The code used for nomogram construction is provided in the Appendix A.

These above steps were then repeated using the “IHC” dataset. The variables used for the development of the multivariable logistic equation included age, size of the tumour, palpable characteristics, location, the histological characteristics of the tumour (i.e., grade, differentiation, necrosis, mitoses and mitotic rate), as well as the immunostaining scores for VEGF and decorin.

### 2.5. Statistical Validation of the Nomograms

The performance of both the “Clinical_train” and the “IHC” nomogram was assessed by determining the concordance index (C-index). The C-index is a measure of goodness of fit for binary outcomes in a logistic regression model and gives the probability of whether the predicted outcome agrees with the observed outcome. The difference between these two measures is Somer’s D (Dxy) value. The C-index was calculated from Dxy using the following formula: C-index = 0.5 × (Dxy + 1).

With nomogram development, it is common practice to use resampling methods to enable validation of the predictive performance of the Cox model used in the nomogram. For this study, the Bootstrap method was employed, with the model iteratively applied to 200 randomly created datasets using cases selected from the original cohort [31]. The results generated by the ‘rms’ validate function in ‘R’ compares the predictive ability of the original data with the mean of those derived by bootstrapping. The difference between the original C-index and the average derived by bootstrapping is an estimate of the overfit or optimism.

### 2.6. Validation of the Nomograms Using an Independent Dataset

The performance of the nomogram was next assessed by generating the C-index using the independent dataset “Clinical_valid”. The bootstrap method was again employed, with the model iteratively applied to 200 randomly selected samples from the independent cohort. The C-index was calculated from Dxy, using the formula (as above).

### 2.7. Nomogram Validation by Manual Calculation of Values

Following the creation of the nomogram, the probability of outcome was manually calculated for each case in the original “Clinical” and “IHC” datasets. Previously excluded cases from the original population of 350 soft tissue sarcoma were included if their “unknown” variable was not required in the nomogram calculation. For the Clinical dataset, this enabled the addition of another 62 cases where ‘size’ had been classified as unknown; the final cohort available for manual validation of the Clinical nomogram was now 232 cases. No additional cases were included in the IHC dataset for manual validation of the IHC nomogram.

### 2.8. Sensitivity, Specificity and ROC Validation of the Nomograms

The probability score for predicted tumour recurrence derived from the nomogram was then plotted against actual tumour recurrence in a ROC curve. Using co-ordinates from the ROC curve, a cut-off value for 3-year local recurrence probability was determined. This cut-off value was then applied to the local recurrence probability that had been determined for all patients in both the “Clinical” and the “IHC” datasets. This enabled a binary recurrence outcome to be predicted. By comparing this predicted outcome with the actual outcome in a 2 × 2 table, the sensitivity, specificity, positive predictive, and negative predictive values could be calculated for both the “Clinical” and the “IHC” nomograms. The area under the curve (AUC) of the ROC curve line was also calculated and compared with the C-index generated by the statistical method described above.

## 3. Results

### 3.1. Clinical Train Dataset

#### 3.1.1. Patient Demographics

The “Clinical_train” dataset contained a total of 102 patients. During the study period, tumour recurrence occurred in 27 patients (27%), with a median DFI of 557 days (range 28–1068 days). From Kaplan–Meier analysis, the palpable characteristics of the tumour (fixed vs. mobile) and various histological characteristics (necrosis, mitotic rate, and grade) were all found to have a significant influence on recurrence. 

Calculated hazard ratios for each individual clinical parameter were determined by univariate Cox regression analysis. These results suggested that a fixed tumour was 4.4 times more likely to recur than a discrete, mobile tumour; a high-grade tumour was 2.6 times more likely to recur than a low-grade tumour; and a tumour with a mitotic index of 3 was 1.9 times more likely to recur than a tumour with a mitotic index of 1 (Appendix A).

Based on the ROC curves generated for each clinical parameter, the predictive ability to determine the actual outcome for patients was considered to be poor for tumour size, differentiation, mitotic rate, necrosis and age; the AUC for these variables was calculated to be between 0.49 and 0.60. Only the variables “Palpable characteristics” and “Grade” showed some ability to distinguish patients, with an AUC of 0.68 and 0.67, respectively (Table 1).

Using co-ordinates from the ROC curves, the cut-off values for “palpable characteristics” and “grade” was determined to be “fixed, immobile” and “grade 2 or grade 3” tumours, respectively. When this predicted outcome was compared to the actual outcome, the following results were obtained:Palpable characteristics: A true positive result was obtained in 21 patients, but a further 32 patients were wrongly predicted to experience recurrence when they did not (i.e., false positive). An accurate prediction of no recurrence was made in 43 patients (i.e., true negative), but tumours recurred in six patients when the test results suggested they would not (i.e., false negative). Overall, this gave a sensitivity of 78%, a specificity of 57%, a positive predictive value of 40%, and a negative predictive value of 88%;Grade: A true positive result was obtained in 15 patients, but a further 17 patients were wrongly predicted to experience recurrence when they did not. An accurate prediction of no recurrence was made in 58 patients, but tumours recurred in 12 patients when the test results suggested they would not. Overall, this gave a sensitivity of 56%, a specificity of 77%, a positive predictive value of 47%, and a negative predictive value of 83%.

#### 3.1.2. Nomogram Construction: Clinical

Using backward selection multi-variable Cox regression analysis, the optimal variables for use in the nomogram was determined, as shown in Table 2. 

Based on these results, “Palpable characteristic”, “Mitotic Rate”, and “Necrosis” were used to generate a nomogram to calculate the probability of being tumour free at 3 years (Figure 1).

### 3.2. Statistical Validation of the Clinical Nomogram

The validation of the Cox model using the training dataset (Clinical_train) generated a Dxy value of 0.45, which equated to a C-index of 73%. With bootstrapping, the Dxy value was 0.44, which equated to a C-index of 72%. From these values, the optimism-corrected estimate of Dxy was 0.4, giving a C-index of 70%.

When the validation of the Cox model was performed using the independent dataset (Clinical_valid), the Dxy value was 0.23, which equated to a C-index of 61%. With bootstrapping, the Dxy value was 0.14, which equated to a C-index of 57%. From these values, the optimism-corrected estimate of Dxy was 0.03, equating to a C-index of 51%.

### 3.3. Manual Validation of the Clinical Nomogram

Using the nomogram is relatively simple and involves three separate steps (Figure 2). Firstly, using the scale for each variable, the ‘Points’ scale at the top of the chart is used to determine the individual value for the relevant characteristic of a patient’s STS. Next, the ‘total score’ of all variables is totalled. Finally, the ‘Total points’ scale is used to determine the ‘probability of outcome’, with values read from the 3-year DFS (disease-free survival) probability scale. 

When using the probability values generated from the nomogram for each case in the “Clinical” database, the resulting ROC curve gave an AUC of 0.67 (95% CI 0.6–0.75, *p* ≤ 0.0001) (Figure 3).

By using the co-ordinates of the ROC curve, the optimal cut-off value of probability to provide a binary predictor of tumour recurrence within 3 years was determined to be >85%. When this value was applied to all cases in the clinical dataset, the nomogram was found to have correctly identified 41 patients where recurrence occurred (true positive), but incorrectly predicted recurrence in 110 patients when no recurrence was observed (false positive). The nomogram accurately predicted tumour-free survival in 73 patients (true negative) but failed to predict recurrence in nine patients (false negative). Overall, the sensitivity, specificity, positive predictive, and negative predictive values for the clinical nomogram were 82%, 40%, 27%, and 89%, respectively.

### 3.4. IHC Dataset

#### 3.4.1. Patient Demographics

The IHC dataset contained a total of 82 patients (Appendix A). Tumour recurrence developed in 26 patients (32%), with a median DFI of 655 days (range 28–1098 days). From the Kaplan–Meier analysis, the immunostaining of VEGF, necrosis, and the palpable characteristics of the tumour were all found to be influential on recurrence. 

The calculated hazard ratios for each individual clinical parameter, as determined by univariate Cox regression analysis, are shown in Table 3. These results suggested that a tumour with diffuse immunostaining for VEGF was 8.4 times more likely to recur than one with low immunostaining. A tumour with >50% necrosis was 7.2 times more likely to recur than one with minimal necrosis, and a fixed tumour was 2.7 times more likely to recur than a mobile one.

Using the ROC curve, the predictive ability of individual test characteristics to reliably determine the actual outcome for patients was considered to be poor (Table 3). For the variables decorin, differentiation, mitotic rate, necrosis, grade, age, and tumour size, the AUC was calculated to be between 0.49 and 0.64. Only VEGF showed some ability to distinguish patients, with an AUC of 0.79.

When using the co-ordinates from the ROC curves, the cut-off value for VEGF to determine a binary decision for recurrence was “1”. When this predicted outcome was compared to the actual outcome, true positive results were obtained in 22 (27%) patients, but a further 17 (21%) patients were wrongly predicted to experience recurrence when they did not (false positive). The accurate prediction of no recurrence was made in 39 (48%) patients (true negative), but tumours recurred in 4 (5%) patients when the test results suggested it would not (false negative). Overall, this gave a sensitivity of 84%, a specificity of 70%, a positive predictive value of 56%, and a negative predictive value of 90%.

#### 3.4.2. Nomogram Construction: IHC 

The stepwise determination of the optimal variables using backward selection multi-variable Cox analysis is shown in Table 4. Based on these results, four variables: VEGF, decorin, mitotic rate, and age, were used to generate a nomogram to calculate the probability of being tumour-free at 3 years (Figure 4).

The validation of the Cox model using all of the cases in the IHC dataset generated a Dxy value of 0. This equated to a C-index of 80%. With bootstrapping, the D-value was 0.61, which equated to a C-index of 81%. This provided an optimism-corrected C-index of 76%.

#### 3.4.3. Manual Validation of the IHC Nomogram

When using the probability values generated from the nomogram for each case in the IHC database, the resulting ROC curve gave an AUC of 0.84 (95% CI 0.76–0.93, *p* ≤ 0.0001) (Figure 5).

When using the co-ordinates of this ROC curve, the optimal cut-off value of probability to provide a binary predictor of tumour recurrence within 3 years was determined to be >90%. When this value was applied to all cases in the IHC dataset, the nomogram was found to have correctly identified 25 patients where recurrence occurred (true positive) but incorrectly predicted recurrence in 31 patients when no recurrence was observed (false positive). The nomogram accurately predicted tumour-free survival in 25 patients (true negative) but failed to predict recurrence in 1 patient (false negative). Overall, the sensitivity, specificity, positive predictive, and negative predictive values of the IHC nomogram were 96%, 45%, 45%, and 96%, respectively.

### 3.5. Summary of Results

When the predictive abilities of individual tumour characteristics were compared with the results of both the clinical and IHC nomogram, the IHC nomogram showed clear superiority in providing a reliable prediction of outcomes, with an AUC of 0.84 (Table 5).

## 4. Discussion

The results from this study suggest that a nomogram may be useful to help predict the likelihood of a canine STS recurring after surgical excision. Of the two nomograms developed in the current study, the inclusion of the immunohistochemical staining characteristics compared to conventional histopathologic characteristics improved the reliability of the prediction provided by the model. While the use of various clinical and histological characteristics of the tumour has been used for many years to help predict potential tumour behaviour [2], this is the first time the use of a graphical calculating tool, such as a nomogram, has been described in veterinary medicine. 

An important attribute of any diagnostic test is its ability to provide an accurate prediction of the true disease status of an individual patient. When individual tumour characteristics, such as size, age, mitotic rate, and necrosis, were used to determine the risk of recurrence, the ability to predict which individual was likely to have an undesirable outcome was not much better than flipping a coin. Only the grade and palpable characteristics of the tumour provided some improved differentiation, but a high degree of uncertainty remained in the prediction. When using these criteria alone, it would be challenging for a clinician to recommend that a dog undergo further treatment when there is up to a 50% chance that the dog has been falsely identified as being ‘at-risk’ and recurrence may actually never occur.

The purpose of the nomograms developed in this study was to identify dogs whose tumours were more likely to recur after surgery. This endpoint was selected as it is known that the local recurrence of the tumour is the most common cause of tumour-related death [2]. If these dogs could be identified earlier, it is possible that their lives could have been saved or prolonged by performing a wider resection of the tumour scar or by providing other adjuvant therapies such as chemotherapy or radiotherapy to prevent the progression of their tumour [3,4,32]. When several characteristics of the tumour, including palpable characteristics, mitotic rate, and necrosis score, were combined into a nomogram using statistical modelling, the ability to predict outcomes improved with a sensitivity of 82%. However, because specificity remained poor, there were almost three dogs wrongly suspected of being at risk of recurrence for every dog correctly identified.

When the immunohistochemical characteristics of the tumour were included in the model, the predictive abilities of the nomogram began to demonstrate some degree of clinical utility. However, even in this instance, there was still an almost 40% false positive rate. This would again create challenges for a clinician who needs to decide whether to recommend additional treatment for an individual patient.

Although the nomograms developed in this current study may not, in their existing form, provide a clinician with the precision required to accurately identify patients where recurrence was more likely, the high sensitivity of the IHC nomogram did more accurately identify patients where recurrence is unlikely to occur. Using the IHC dataset, the nomogram accurately predicted tumour-free survival in more than 96% of patients. Within the original study population, the risk of recurrence was almost 30%, and when using individual tumour characteristics, there was no reliable ability to distinguish the patients according to if recurrence was likely or unlikely to occur. However, by using the information from the IHC nomogram, a clinician could confidently identify the patients where tumour recurrence would not occur. For the owners of these dogs, progressing from a 30% possibility that recurrence could develop after surgery to an almost 100% certainty that their dog’s tumour was not going to recur could provide a tremendous degree of relief.

The inability of the nomograms developed in this study to reliably predict which STS would recur is a major weakness and suggests they lack some vital distinguishing characteristics that would improve the differentiation. One of the obvious deficiencies in the data used to develop the nomograms in the current study is the absence of information on the completeness of tumour resection or the histological margin. It is generally accepted that the demonstration of a resection margin that is clear of tumour cells is considered the best predictor for improved local tumour control [2,10,33,34,35,36,37,38,39]. It is, therefore, likely that if the information on the histological completeness of the surgical margin were included in the nomogram, this would improve the specificity of the nomograms developed in this study.

Despite this lack of information on the surgical margin, it is interesting that the nomograms developed in this study were able to provide a reasonably accurate prediction of tumour recurrence. This lends support to the hypothesis that the status of the surgical margin is not always a definitive guide to a patient’s outcome after surgery, with other aspects of tumour biology influencing the ability for a tumour to regrow after surgery, irrespective of whether the surgeon has successfully removed all of the neoplastic cells. It is recognised that STSs may recur even when the histologic margins have been determined to be complete, and an incomplete surgical margin does not mean tumour recurrence is inevitable. In dogs, recurrence rates for STSs of between 5–22% have been reported when a clean resection has been achieved, and no regrowth may occur in up to 83% of patients when incomplete or close resection margins have been described [10,40]. Similar findings have been reported for human soft tissue sarcoma [41]. 

There are many limitations to the nomograms developed in the current study that would limit their immediate application in clinical practice. The first limitation of the proposed nomograms is in the construction of the algorithm that resides behind the pictorial nomogram. The nomograms described in the current study were constructed using data from a retrospective study that assessed the outcome for dogs with STSs that were surgically excised in first-opinion practice [26]. The data on which the nomograms were constructed was collected retrospectively; this may result in recall bias or inaccuracy within the responses. Veterinarians completing the survey were reliant on clinical notes that had been written many years previously. This raises the possibility that some of the clinical information supplied about the tumour may be inaccurate. This deficiency could have an impact on the clinical nomogram, which utilised a subjective description of the tumour in its algorithm. For example, the distinction of whether a tumour is “fixed” or “mobile” is subject to individual interpretation by the clinician. In the IHC nomogram, the variables used were less liable to misinterpretation, as it utilised more objective or defined data, such as age, mitotic rate, and the immunostaining characteristics of VEGF and decorin.

An additional deficiency of the current study was the small number of cases used to construct the nomogram. The small size of the population cohorts used in both the clinical and the IHC nomogram will have a significant impact on the ability to detect statistical differences between the covariates selected for inclusion within the nomogram. Most human studies where nomograms have been described and accepted within the clinical community have typically utilised sample sizes 10–100 times larger than that used in the current study. Because of the small number of cases in the cohort, the covariates were selected when the significance was only 0.15 rather than a more conventional figure of 0.05. The use of 0.15 means that there is a 15% (almost one in six) chance that the selected variable does not actually influence the outcome as suspected. In contrast, when using a *p*-value of 0.05, this means that there is just a 5% (1 in 20) chance that the effect of the variable is simply due to chance. By broadening the inclusion of the potentially relevant cases into a selected variable in this way, the accuracy of the nomogram will suffer. Because the selected variable may now lack sufficient distinguishing power, the nomogram may identify cases that are at risk of developing tumour recurrence when in fact, they did not. This lack of accuracy will increase the number of false-positive results and may explain the poor specificity of the nomograms developed in this study. 

Another potential source of error that can limit the reliability of the nomogram is if any of the selected variables are likely to exert an influence on each other. If the variables used in the nomogram are not truly independent of each other, then there are no additional benefits from including the additional characteristic in the algorithm. This dependence may also bias the selection of cases, as a case with one dependent variable is likely to gain an additional score on the nomogram from its related variable. In the IHC nomogram developed in the current study, the variables decorin, mitotic rate, and VEGF were identified by the Cox model as having an independent influence on the outcome and were selected as characteristics to be used in the nomogram. However, at a physiological level, decorin is recognised as an important tumour suppressor [42]. It follows that reduced levels of decorin within a tumour will increase the availability of VEGF and other sequestered cytokines within the tumour microenvironment [43]. The varied bioavailability of these cytokines within the tumour microenvironment will likely have diverse consequences on the tumour, including influences on cellular metabolism, mitotic activity, and the production of other unmeasured molecules that may influence tumour progression. It follows that the true independence of VEGF, decorin, and mitotic index cannot be assured, and it is likely that more sophisticated statistical tools would be required to analyse this further.

Finally, the gold standard for nomogram calibration is to utilise an independent dataset, i.e., one that is distinct from the population originally used to develop the nomogram [15]. In the current study, external validation was performed for the clinical nomogram by splitting the original dataset into two populations, with one set used for the development and training of the model and the other for external validation. It should be noted that splitting the original population into two, as was performed in this current study, does not create a truly independent dataset. This is because the population used for the validation has ultimately been derived from the same study as the training dataset. The cases in the validation dataset are, thus, influenced by the same biases and limitations that affected the training dataset; these biases and limitations were outlined in the previous section. Confidence in the performance of any nomogram will only be achieved when it has been validated against an external population.

## 5. Conclusions

Evidence from this study suggests a nomogram could play an important role in helping to identify patients who either have no risk of recurrence after surgery or who are liable to experience recurrence at some time in the future. These latter patients may choose to undergo additional therapy—either a wider surgical resection, radiation therapy, or chemotherapy—to help reduce this risk of recurrence. 

The current study provides the first evidence in veterinary oncology to support the potential role of the nomogram to assist in predicting the outcome for patients after surgery for STS. From the evaluations performed, a nomogram that incorporates data from an IHC interrogation of the tumour is more reliable than a nomogram that does not. However, while it is evident that nomograms may have the power to become an important component of decision-making for the cancer patient, they will need to demonstrate robust reliability and accuracy if they are to completely supplant the insight and judgement of a clinical expert. Additional study will be required to ensure that such a tool can be reliably and confidently incorporated into routine surgical planning.

## Figures and Tables

**Figure 1 vetsci-10-00266-f001:**
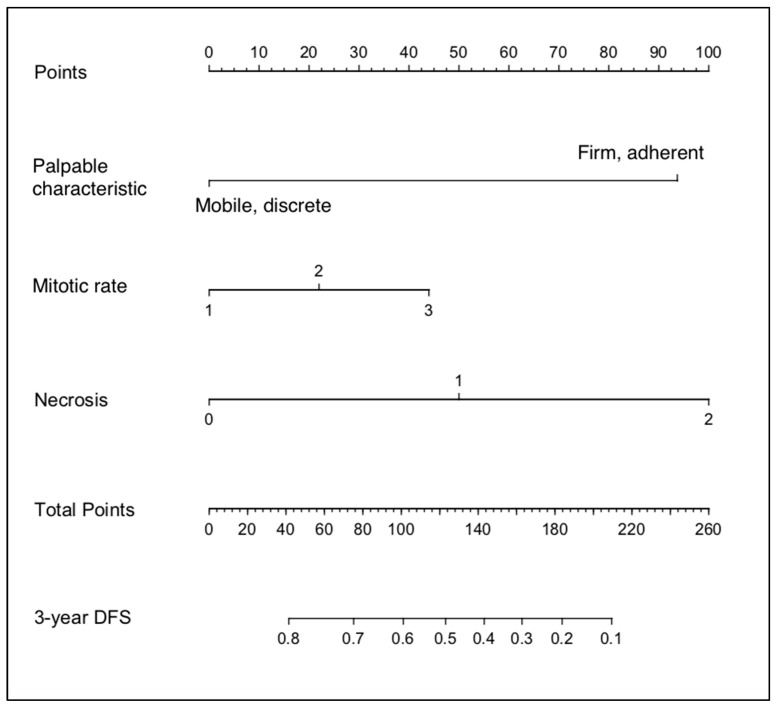
Nomogram developed from the Clinical_train dataset.

**Figure 2 vetsci-10-00266-f002:**
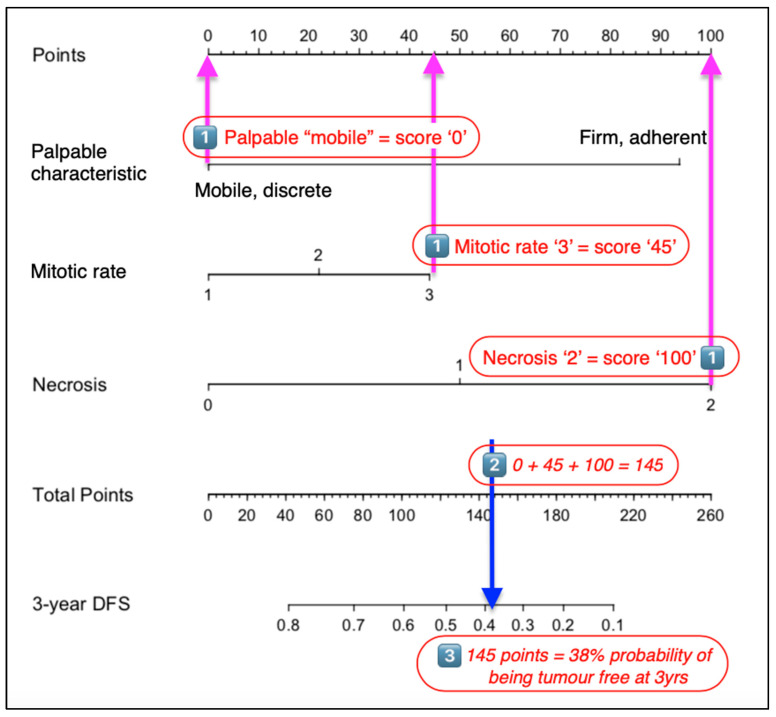
Steps to using a nomogram. (1) Determine the POINTS scored for each characteristic defined in the nomogram. (2) Total these points and identify this value on the TOTAL POINTS scale. (3) The 3-year disease-free interval is then determined using the proportional scale that is in line with the value identified in the previous calculation.

**Figure 3 vetsci-10-00266-f003:**
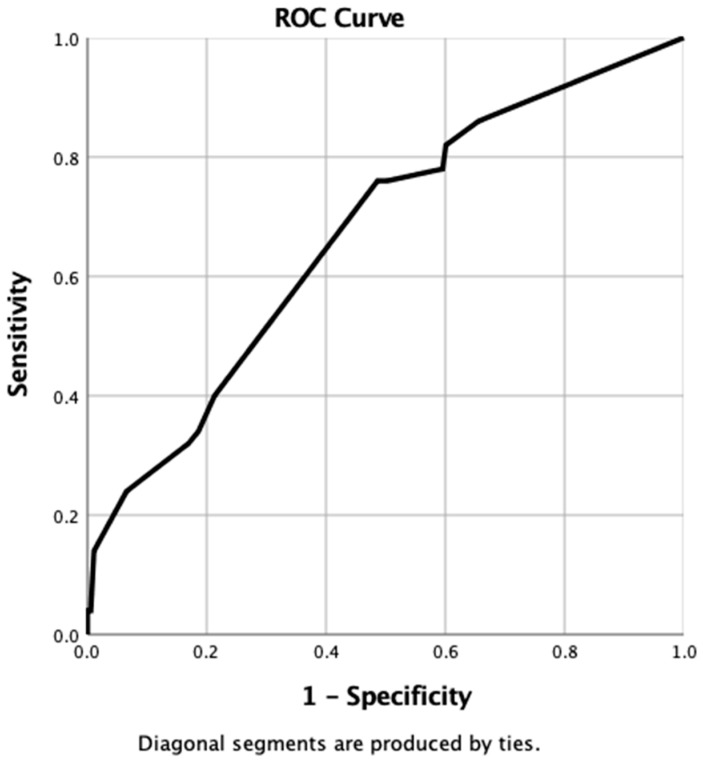
ROC curve generated from probabilities derived from clinical nomogram.

**Figure 4 vetsci-10-00266-f004:**
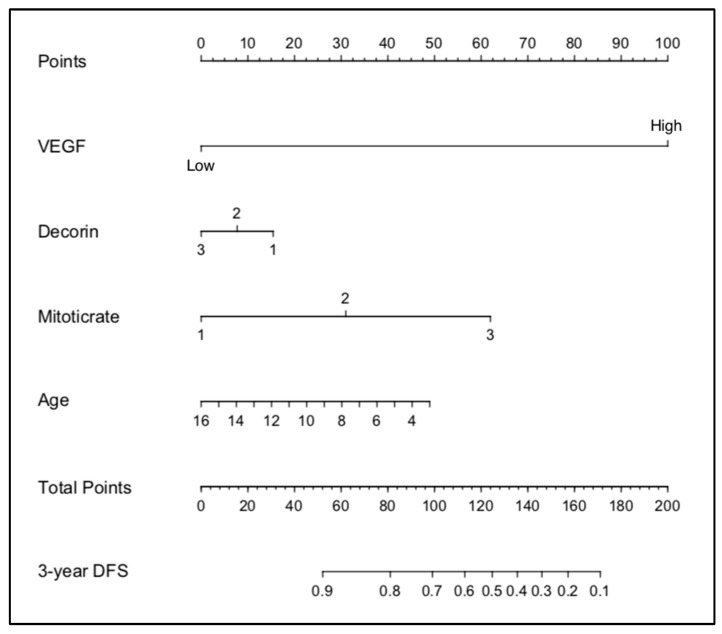
Nomogram developed from the IHC dataset.

**Figure 5 vetsci-10-00266-f005:**
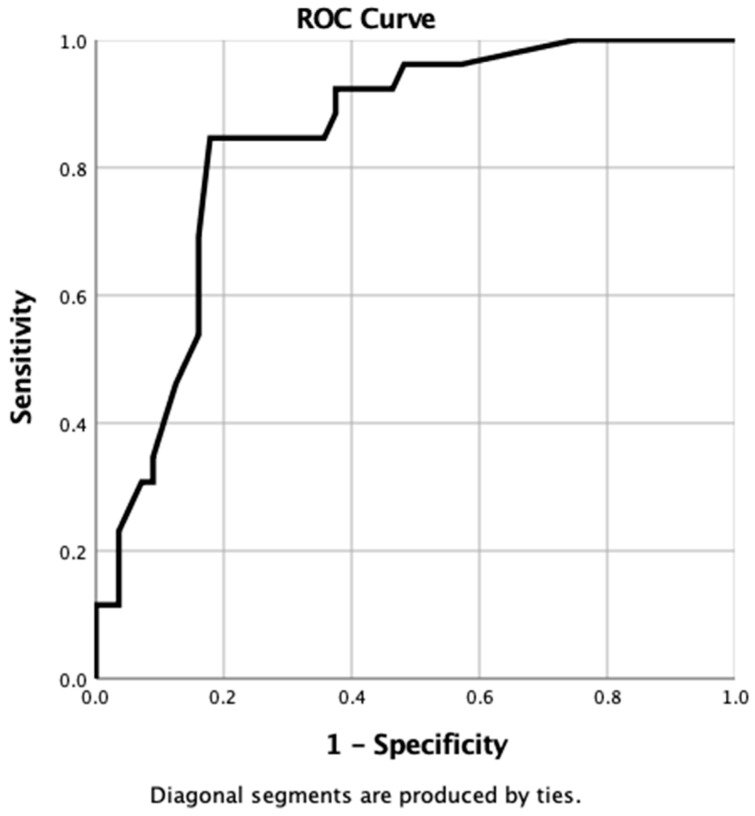
ROC curve generated from probabilities derived from IHC nomogram.

**Table 1 vetsci-10-00266-t001:** AUC of ROC curve for individual parameters in the Clinical_train dataset.

Parameter	AUC	Significance	95% Confidence Interval
Size	0.581	*p* = 0.215	0.45–0.711
Palpable characteristics	0.676	*p* = 0.007	0.561–0.79
Differentiation	0.534	*p* = 0.606	0.406–0.661
Mitotic rate	0.584	*p* = 0.197	0.453–0.715
Necrosis	0.604	*p* = 0.109	0.472–0.737
Grade	0.666	*p* = 0.011	0.541–0.792
Age	0.488	*p* = 0.856	0.376–0.6
Mitoses	0.629	*p* = 0.047	0.501–0.757

**Table 2 vetsci-10-00266-t002:** Multivariable COX regression analysis on the Clinical_train database to identify the characteristics to be used in the nomogram.

	Clinical Characteristic	Significance (*p* Value)	HR	95.0% CI for HR Lower Upper	
1	Well-differentiated	0.995	-	-	-
	Moderately differentiated	0.949	0.967	0.346	2.7
	Poorly differentiated	0.966	1.054	0.096	11.595
2	Size (<1 cm)	0.778	-	-	-
	Size (1–5 cm)	0.329	1.623	0.613	4.296
	Size (>5 cm)	0.611	1.334	0.44	4.048
3	Age	0.79	0.98	0.846	1.136
4	Mitoses	0.731	1.011	0.95	1.075
5	Grade 1	0.657	-	-	-
	Grade 2	0.736	0.783	0.188	3.253
	Grade 3	0.389	0.303	0.02	4.598
6	Palpable (discrete)				
	Palpable (firm, immobile)	0.035	2.403	1.065	5.421
	Mitotic rate score 1	0.015	-	-	-
	Mitotic rate score 2	0.11	2.141	0.841	5.446
	Mitotic rate score 3	0.007	5.08	1.571	1422
	Necrosis score 1	0.181	-	-	-
	Necrosis score 2	0.156	0.49	0.183	1.313
	Necrosis score 3	0.318	2.128	0.483	9.377

**Table 3 vetsci-10-00266-t003:** AUC of ROC curve for individual parameters in the IHC dataset.

Parameter	AUC	Significance	95% Confidence Interval
VEGF	0.786	*p* ≤ 0.001	0.677–0.895
Decorin	0.534	*p* = 0.628	0.398–0.669
Differentiation	0.488	*p* = 0.863	0.354–0.622
Mitotic rate	0.517	*p* = 0.804	0.379–0.655
Necrosis	0.54	*p* = 0.572	0.399–0.68
Grade	0.506	*p* = 0.936	0.37–0.642
Age	0.626	*p* = 0.072	0.5–0.752
Size	0.52	*p* = 0.779	−0.665

**Table 4 vetsci-10-00266-t004:** The stepwise backward selection of variables in the IHC database using Cox regression analysis identified four characteristics of appropriate significance to be used in the nomogram.

	Clinical Characteristic	Significance	HR	95.0% CI for HR	
		(*p* Value)		Lower	Upper
1	Well-differentiated	0.9	-	-	6
	Moderately differentiated	0.7	1.3	0.314	
	Poorly differentiated	0.9	0.6	0.004	
2	Size (<1 cm)	0.8	-	-	-
	Size (1–5 cm)	0.3	1.623	0.613	4.296
	Size (>5 cm)	0.6	1.334	0.44	4.048
3	Grade 1	0.6	-	-	-
	Grade 2	0.7	0.723	0.135	3.882
	Grade 3	0.3	0.231	0.014	3.786
4	Necrosis score 1	0.3	-	-	-
	Necrosis score 2	0.3	0.564	0.198	1.607
	Necrosis score 3	0.4	1.842	0.437	7.764
5	Palpable (discrete)		-	-	-
	Palpable (firm, immobile)	0.2	1.769	0.675	4.635
6	VEGF low		-	-	-
	VEGF high	<0.0001	31.25	5.197	187.903
	Decorin type 1	0.1	-	-	-
	Decorin type 2	0.9	1.097	0.394	3.06
	Decorin type 3	0.1	0.397	0.134	1.18
	Mitotic rate score 1	0.01	-	-	-
	Mitotic rate score 2	0.6	0.727	0.207	2.551
	Mitotic rate score 3	0.002	25.271	3.257	19062
	Age	0.1	0.856	0.71	1.031

**Table 5 vetsci-10-00266-t005:** Comparison of predictive abilities of individual tumour characteristics were compared with the results of both the clinical and IHC nomograms.

	C-Index	AUC of ROC Curve (95% CI)	Sens	Spec	PPV	NPV
**Individual Characteristics**						
Palpable		0.68	78%	57%	40%	88%
Grade only		0.67	56%	77%	47%	83%
VEGF		0.79	84%	70%	56%	90%
**Nomograms**						
Clinical nomogram						
Training dataset	71%	0.67 (0.6–0.75)	82%	40%	27%	89%
Validation dataset	51%					
IHC nomogram	75%	0.84 (0.76–0.93)	96%	45%	45%	96%

## Data Availability

The data presented in this study are available in Appendix A.

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
