# Peer review of "Development of a Nomogram to Predict the Outcome for Patients with Soft Tissue Sarcoma"

_vetsci, 2023, doi:10.3390/vetsci10040266_

Round 1

Reviewer 1 Report

Dear Authors, 

in my opinion, this is very, very interesting manuscript. But I have a few questions. Maybe not all is very clear to me. The first question is about Tables 2 and 5. Why in the tables are legible only fields with results? In my version, the fields with data are totally black. Very interesting is a part of a discussion about the margins of surgically removed tissues with STS. Maybe this is important data for the nomogram too? Were margins complete or incomplete? Maybe with results of IHC can be very helpful. The next question is about IHC. Maybe I do not understand, but why only two antibodies VEGF and decorin? Maybe next to the mitotic index in HE staining, the Ki67 antibody in IHC can be very helpful ? The suggestion from the Authors that the data are retrospective is very important. I have an idea for the next study performed on the questionnaire for veterinarian surgeons and pathologists only with currently treated and diagnosed STS- dogs?  The idea of nomograms for oncological patients in veterinary medicine is very good and important. Maybe in this age, the idea of using Artificial Intelligence to perform nomograms can be very good?

Regards

Reviewer 2 Report

Dear Authors, 

I really appreciated the novelty and usefulness of your work, even if in M&M is quite hard to understand for a clinician. 

Minor check are needed:

- check the references, proper position in the phrase.

- please add the simple summary

- please make table readable and add a legend, not only a title

Reviewer 3 Report

My compliments to the authors as I really enjoyed the article and find it will be useful in the veterinary medical community. Having some knowledge of statistics I enjoyed many parts.
I just have a few small suggestions to make the article better.

-in the abstract specify what AUC means

- line 28 close the parenthesis

- line 83 not everyone knows how a nomogram works I would write a few lines on how a nomogram is used

- Table 1 for me the R script is too specific and can be put as supplementary material, it weighs down the reading and confuses the reader

- I would also put the 2 datasets in the supplementary materials

-table 4 I would replace clinical characteristic with clinical and histopathological characteristic

- Figura 1. what do you think about putting an example of a case after figure 1, giving the characteristics of the case and getting the result on the nomogram?
